# Ontology-Based Approach Supporting Multi-Objective Holistic Decision Making for Energy Pile System

**Kun Meng** [1,2], **Chunyi Cui** [2,*], **Haijiang Li** [3] **and Hailong Liu** [2]

1 College of Transportation, Shandong University of Science and Technology, Qingdao 266590, China; skd996565@sdust.edu.cn
2 Department of Civil Engineering, Dalian Maritime University, Dalian 116026, China; liuhailong@dlmu.edu.cn
3 Cardiff School of Engineering, Cardiff University, Queen's Buildings, The Parade, Cardiff CF24 3AA, Wales, UK; Lih@cardiff.ac.uk
* Correspondence: cuichunyi@dlmu.edu.cn

**Abstract:** The traditional way of designing energy pile system is mostly single domain/objective oriented, which lacks of means to coherently consider different while relevant factors across domains. The cost for life cycle design, construction and maintenance, return of investment, $CO_2$ emission related sustainable requirements, and so on also need to be considered, in a systematic manner, along with the main functional design objective for loading capacity and robustness. This paper presents a novel multi-objective holistic approach for energy pile system design using ontology based multi-domain knowledge orchestration, which can holistically provide the designers with across domain factors regarding financial, safety, and environmental impact, for smart and holistic consideration during the early design stage. A prototypical ontology-based decision tool has been developed, aiming at the holistic optimization for energy pile system by combining ontology and Semantic Web Rule Language rules. A case study was performed to illustrate the details on how to apply knowledge query to provide a series of design alternatives autonomously by taking different design parameters into account. The method has demonstrated its practicability and scientific feasibility, it also shows the potential to be adopted and extended for other domains when dealing with multi-objective holistic design making.

**Keywords:** ontology; holistic design; $CO_2$ emission reduction; energy pile

## 1. Introduction

It is generally acknowledged that a large amount of greenhouse gases, produced by the building industry, has become a significant cause of global warming [1]. In Europe, 75% of the residential energy consumption is used for heating and cooling of buildings [2]. In addition, the global energy demand for the cooling of buildings has increased by 70% due to global warming [3].

As a pioneering work of geothermal heat pump technology, Morino et al. [4] proposed the geothermal energy pile system (EPS), originally using renewable ground source heat to reduce building energy consumption of heating and cooling, in which the heat exchangers are generally set up inside the pile without additional costs of drilling and installing [5], as shown in Figure 1. Such a system transmits heat or cold from ground to buildings during winter or summer, respectively. Due to high heat storage capacity and thermal conductivity of a concrete pile, the geothermal EPS using renewable ground source has great energy efficiency without greenhouse emissions.

Over the last decade, an increasing amount of research has been devoted to the heat exchangers, mechanical performance, and design guidance of EPS. Omer and Haroglu [6] used PLAXIS to investigate the geotechnical characteristics of piles fitted with high-density polyethylene. As for the heat exchangers of EPS, Man et al. [7] considered the heat capacity of the pile and proposed a simplified cylindrical source model and analytical solutions.

Hu et al. [8] further presented a more rigorous cylindrical model to take the radial effect of a large diameter pile into account. Subsequently, Zhang et al. [9] investigated the heat transfer characteristics of an energy pile, with spiral coils based on analytical heat transfer models and experiments. Three dimensional numerical models for heat transfer were also developed to exam the influences of heat exchanger indicators on the thermal efficiency of EPS [10–12]. In addition, some full-scale site investigations and laboratory thermal tests were performed to investigate the influences of the velocity and temperature of circulating water, as well as operation mode on the heat exchange capacity [13–16].

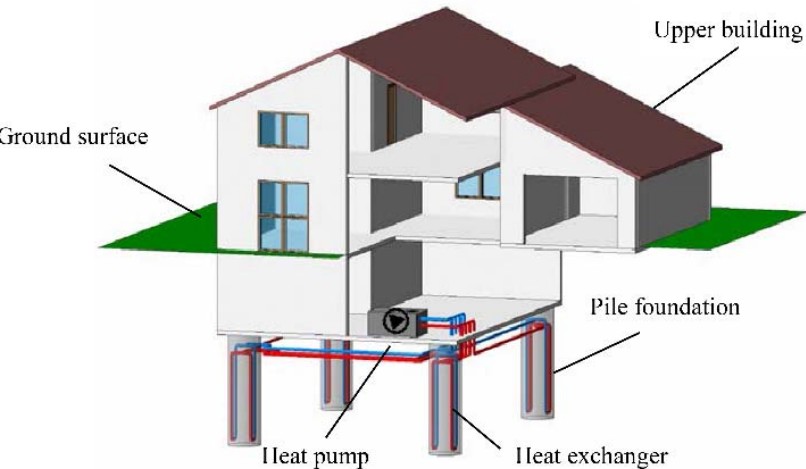

**Figure 1.** The geothermal energy pile system.

In practice, however, temperature changes of the energy pile can cause additional stresses and deformations in the pile-soil system, which affect the serviceability of EPS. Laloui [17] investigated in situ to quantify the thermal effects on the bearing capacity of heat exchanger piles. Subsequently, Binod et al. [18] and Ozudogru et al. [19] investigated the influence of boundary conditions of soil on the thermo-mechanical characteristics of energy piles. Furthermore, Loveridge et al. [20] and Jeong et al. [21] found the primary factors, which influence the heat transfer and thermal–mechanical interactions of energy piles. Chen et al. [22] performed a parametric analysis to reveal the roles of the strength, stiffness, and stress-strain indicators for the pile-soil system, which can provide guidance for preliminary design.

However, most of the previous studies and available standards are related to simplified methods and some specific software for numerical computation, which have obvious limitations to coherently consider different aspects across domains. Actually, the economic feasibility of each geothermal project directly depends on the balance between energy savings and energy costs [23,24]. Therefore, besides fulfilling the main functional design objectives of EPS for loading capacity and heat transfer, the cost for life cycle design, construction and maintenance, return of investment, and $CO_2$ emission related sustainable requirements also need to be considered in a holistic manner.

Ontology, as a novel Semantic Web tool, has been widely adopted in knowledge sharing and exchanging between various domains [25,26]. The primary characteristics of ontology, consisting of semantic structure, machine processing capability, and reasoning function, give a significant tool to conduct a holistic design procedure of modelling various domain knowledge. Yurchyshyna et al. [27] proposed an ontology-based method for formalizing and organizing consistency requirements in construction, according to building codes. Dibleyb et al. [28] developed an ontology framework for intelligent sensor-based building monitoring. Hou et al. [29] proposed an ontology-based method for achieving low embodied energy and carbon during construction design.

Based on the aforementioned literature review, the existing ontology-based methods mainly focus on the construction of buildings that are not suitable for the design of EPS.

Furthermore, the common design methods for EPS primarily consider the main functional design objectives of EPS for loading capacity and heat transfer. Little work, with regard to EPS, has been devoted to holistic design, with consideration of multi-objective (including loading capacity, heat transfer, the cost for life cycle design, construction and maintenance, return of investment, and $CO_2$ emission related sustainable requirements). However, compared with the single-objective (only considering the main functional design objectives of EPS for loading capacity and heat transfer) design method, the multi-objective design method can consider the cost for life cycle design, construction, and maintenance, return of investment, and $CO_2$ emission related sustainable requirements holistically. Therefore, the multi-objective design method can provide more alternative optimization schemes for engineers in a comprehensive insight. The main purpose of this paper is to propose a novel ontology-based approach of leveraging knowledge modeling, for EPS design, to achieve multi-objective design optimization.

Therefore, in light of the above, a specific ontology-based decision support system named OntoEPS is to be built to implement multiple optimized design solutions not only for loading capacity and heat transfer but also minimum embodied carbon and cost recovery period. The structure of this paper is introduced as follows: Section 2 presents the key parameters for the holistic design of EPS, as well as the detailed procedure for the design and development of OntoEPS. Then, a case study of the multi-objective optimization design method is performed in Section 3. Finally, Section 4 presents the primary conclusion of this study.

## 2. Multi-Objective Holistic Design and Development of OntoEPS

### 2.1. Determination of the Key Parameters for Holistic Design of EPS

The primary indicators for the multi-objective holistic design of the EPS consist of equipment cost, cost recovery period, $CO_2$ emission reduction, and the vertical loading capacity. The OntoEPS is to be built to implement multiple optimized design solutions for these multi-objective indicators.

(1)    The equipment cost

The equipment cost mainly consists of the cost of heat exchanger tubes and heat pumps of EPS. The cost of heat exchanger tubes can be obtained by Equation (1)

$$C^{\mathrm{T}} = \sum_{i=1}^{n} C_i^{\mathrm{T}} \times L_i^{\mathrm{T}} \times N_i \tag{1}$$

where $i$ represents the $i^{\mathrm{th}}$ pile type, $L_i^{\mathrm{T}}$ represents the length (m) of heat exchanger tube and $C_i^{\mathrm{T}}$ represents the price of heat exchanger tube (USD) per unit length. $N_i$ and $C^{\mathrm{T}}$ are the number and the total cost (USD) of heat exchanger tube, respectively.

Therefore, the cost of heat pumps is given as

$$C^{\mathrm{P}} = \sum_{j=1}^{m} C_j^{\mathrm{P}} \times N_j^{\mathrm{P}} \tag{2}$$

where $j$ represents the $j^{\mathrm{th}}$ type of heat pump. $C_j^{\mathrm{P}}$ is the price, $N_j^{\mathrm{P}}$ is the number of the heat pump, and $C^{\mathrm{P}}$ represents the total cost.

Then, it gives

$$C^{\mathrm{E}} = C^{\mathrm{T}} + C^{\mathrm{P}} \tag{3}$$

where $C^{\mathrm{E}}$ is the total equipment cost, $C^{\mathrm{T}}$ and $C^{\mathrm{P}}$ represent the costs of heat exchanger tube and heat pump, respectively.

(2)    The cost recovery period

The power of heat exchanged by EPS can be obtained by:

$$H^{\mathrm{T}} = \frac{\sum\limits_{i=1}^{n} H_i \times L_i^{\mathrm{S}} \times N_i}{1000} \tag{4}$$

where $i$ represents the $i^{\mathrm{th}}$ pile type, $H_i$ is the heat exchanged per meter of the $i^{\mathrm{th}}$ pile type (W/m) per hour, which can be determined by the thermal response test (TRT) of the single pile. $L_i^{\mathrm{S}}$ is the length of the $i^{\mathrm{th}}$ type pile. $H^{\mathrm{T}}$ represents the total power of the heat exchanged by the EPS per hour (kWh).

Therefore, the cost recovery period can be further calculated by:

$$T^{\mathrm{C}} = \frac{C^{\mathrm{E}}}{H^{\mathrm{T}} \times P} \tag{5}$$

where $C^{\mathrm{E}}$ is the whole equipment cost (USD), and $H^{\mathrm{T}}$ represents the power of exchanged heat of the EPS per hour (kWh). $P$ and $T^{\mathrm{C}}$ refer to the price of electricity (USD/kWh) and the cost recovery period (hour), respectively.

(3)    The $CO_2$ emission reduction

The $CO_2$ emission reduction refers to the $CO_2$ emissions reduced by EPS during the cost recovery period. According to the relevant research [30], the consumption of per kWh electricity leads to 0.997 kg $CO_2$ emission. The $CO_2$ emission reduction can be calculated by:

$$CO_2^{\mathrm{E}} = H^{\mathrm{T}} \times T^{\mathrm{C}} \times 0.997 \tag{6}$$

where $H^{\mathrm{T}}$ is the power of heat exchanged by the EPS (kWh), $T^{\mathrm{C}}$ is the cost recovery period (hour), and $CO_2^{\mathrm{E}}$ represents the $CO_2$ emission reduction (kg) within the cost recovery period.

(4)    The vertical bearing capability (VBC)

The VBC of EPS can be calculated by:

$$Q = \sum\limits_{i=1}^{n} \frac{Q_i^{\mathrm{uk}}}{K} \times N_i \tag{7}$$

where $i$ represents the $i^{\mathrm{th}}$ pile type, $Q_i^{\mathrm{uk}}$ is the standard value of the ultimate VBC of the $i^{\mathrm{th}}$ pile type (kN), $K$ represents the safety coefficient, and $Q$ represents the VBC of the EPS.

According to these basic mathematical formulations, the implicit form of the multi-objective formulation is to be built by leveraging Ontology.

### 2.2. Design and Development of OntoEPS

#### 2.2.1. The System Framework of OntoEPS

The OntoEPS includes four modules: namely, knowledge base (KB), ontology management system (OMS), rules editor (RE), and query interface (QI), as shown in Figure 2. The KB is the primary essential section of OntoEPS where the basic data can be saved as Ontology Web Language (OWL) files. The OMS provides editors to create and update the ontology model. An open source ontology software, Protégé 5.2, is used to implement the development of OntoEPS. The RE is generally employed to edit and run the Semantic Web Rule Language (SWRL) rules of the ontology model. In addition, the designer can also obtain related results by inputting specific demands from the Semantic Query Web Rule Language (SQWRL) query interface.

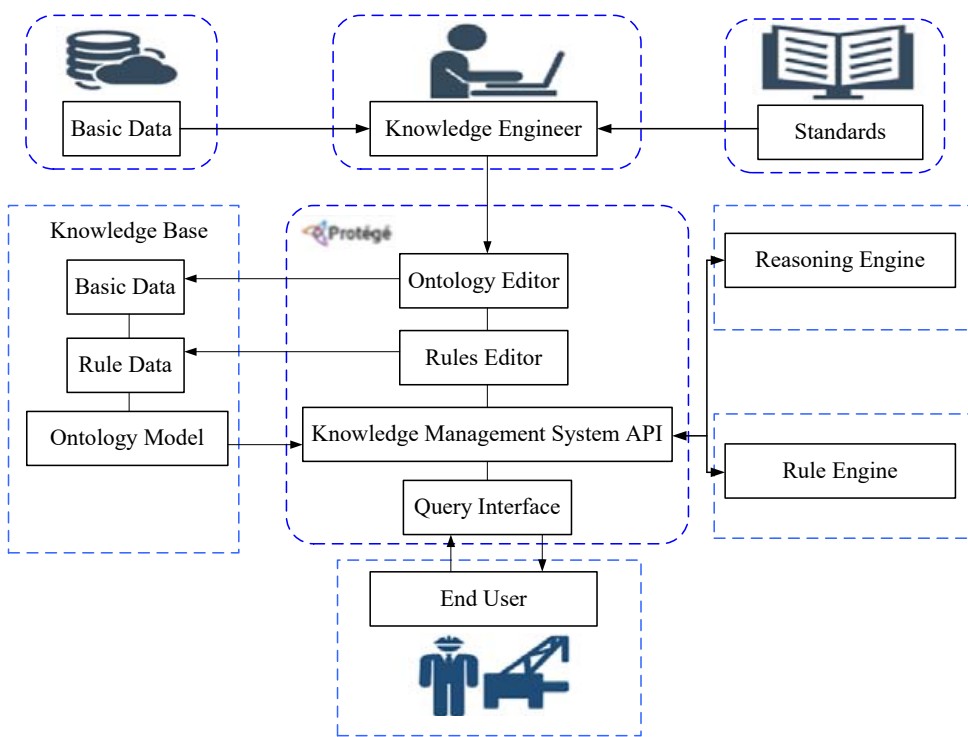

**Figure 2.** The developed framework of OntoEPS system.

## 2.2.2. The Development of OntoEPS System

The Ontology Development 101 [31] is selected as the specific methodology to develop the OntoEPS system. As the most extensively used methodology of ontology development, the Ontology Development 101 provides step-by-step guidance for the implementation of ontology based on Protégé.

Knowledge Identification and Knowledge Specification are the preliminary steps to be prepared for the establishment of the ontology model. In the Knowledge Identification step, the main domains related to OntoEPS are determined according to the primary functional design objectives of EPS and combined with the existing knowledge models for further analysis, whereby all the essential definition of domains are introduced as a constructed glossary. In Knowledge Specification step, a specific description of the knowledge model is established, and then, a semi-formal modelling is further built up by adopting the Unified Modeling Language (UML) language.

In this study, the ontology model for the holistic design of the EPS mainly involves pile foundation engineering, building construction, and engineering of heat and ventilation in terms of safety, economy, and environmental impact.

The organized concepts and terms of this developed OntoEPS system primarily follow the standard with the reference to some existing ontology models [29,32]. The flow diagram of UML classes, for the core concepts of OntoEPS, is shown in Figure 3.

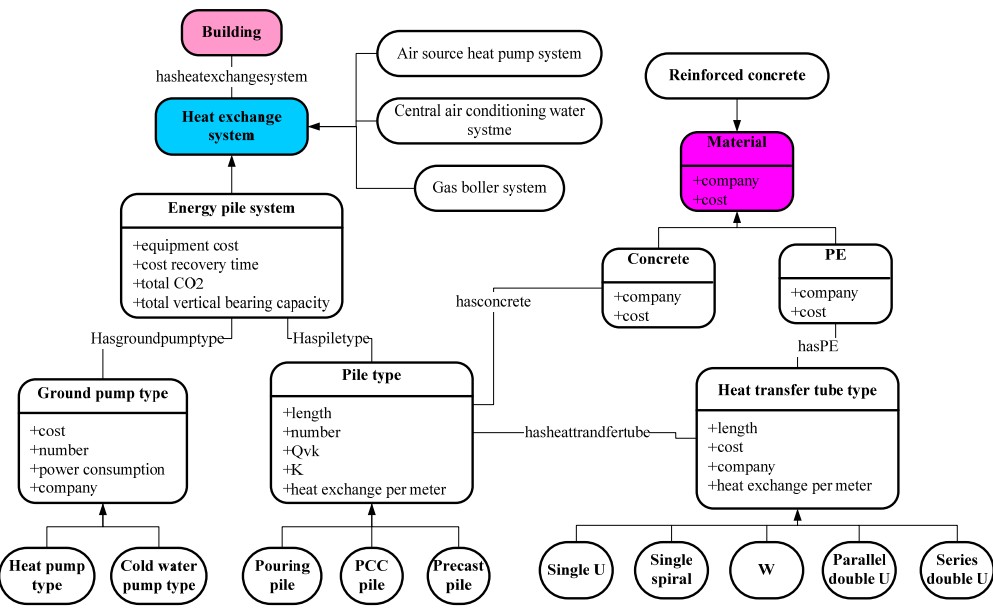

**Figure 3.** Flow diagram of UML classes for the core concepts of OntoEPS.

### 2.2.3. The Procedure of Ontology Development

Figure 4 shows the procedure of ontology development for the EPS, which can be further specified as follows:

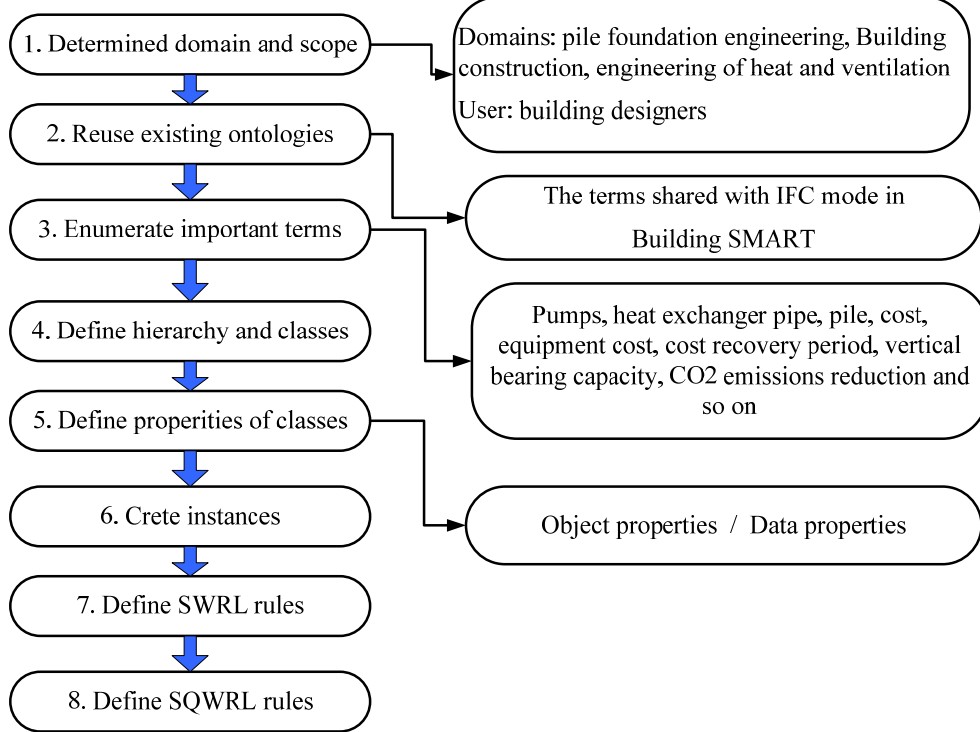

**Figure 4.** Procedure of ontology development for the EPS.

Step 1: the correlative domain and scope of the ontology are determined based on some basic questions (BQ) and competency questions (CQ).

Step 2: IFC mode of the building SMART is used as the primary development standard, which introduces the developed ontologies for the holistic design of EPS [33].

Step 3: a glossary including all the essential terms, in regard to the sustainability, cost, and safety design of EPS, is constructed.

Step 4: according to the essential terms elicited in the previous step, a top-down establishment of the general classes, and corresponding classification of these classes, is conducted [34,35], as shown in Figure 5a.

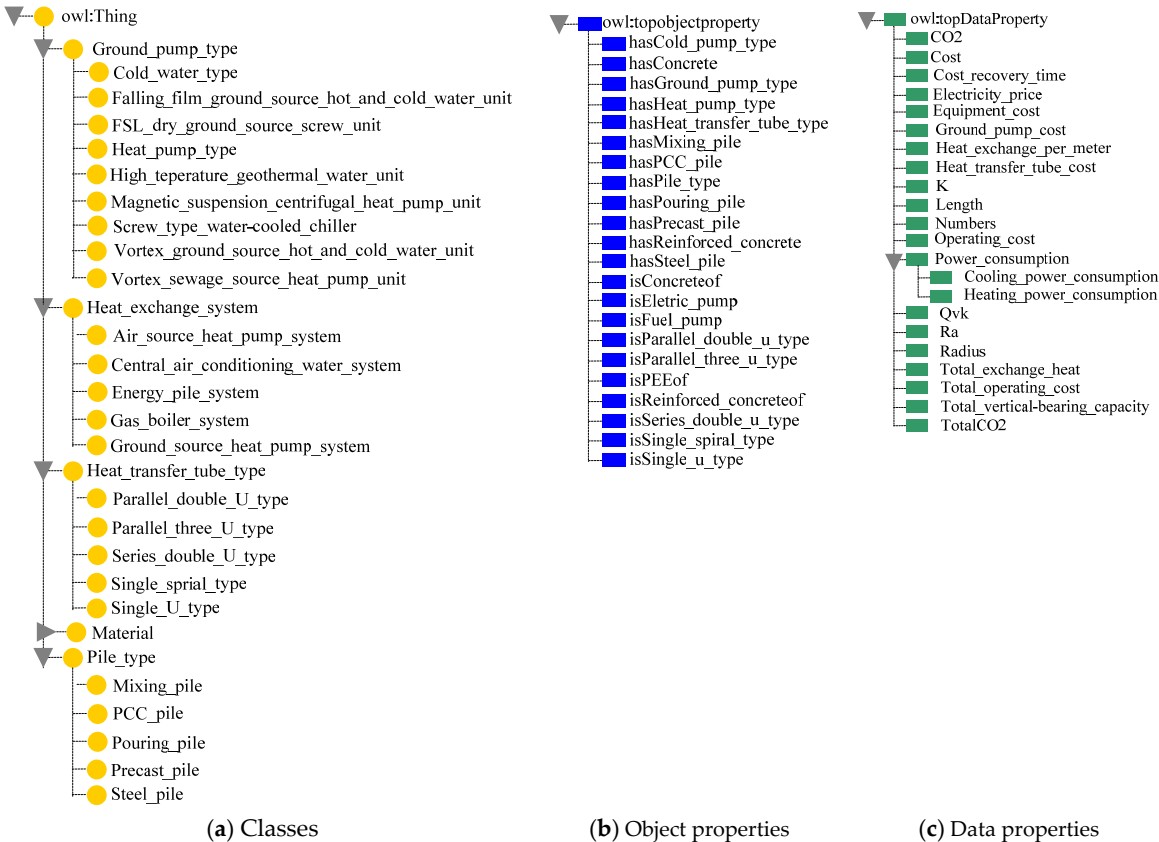

(**a**) Classes · · · · · · · · · · · · (**b**) Object properties · · · · · · · · · · · · (**c**) Data properties

**Figure 5.** The developed ontology in the protégé-OWL 5.2.

Step 5: the built classes have two kinds of properties—namely, object properties and data properties—as shown in Figure 5b,c, respectively. The objective properties describe the relationships between different classes. The data properties denote the features of the specific class instances.

Step 6: A specific instance holding the same hierarchical place with the class 'Energy_pile_system' is created as shown in Figure 6.

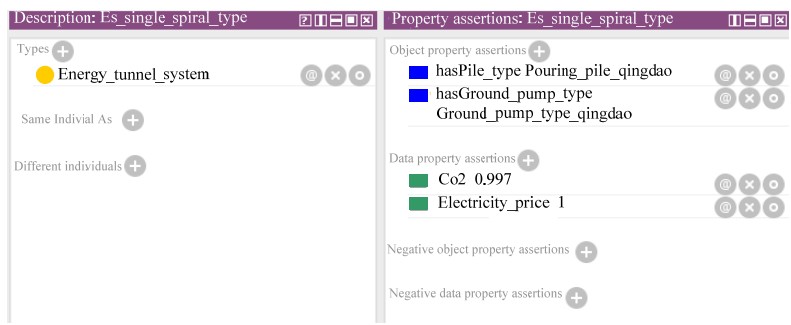

**Figure 6.** Creation of an instance.

Step 7: SWRL rules for holistic design of EPS are performed to increase the adaptability for reasoning of ontology. The atoms of SWRL rules mainly consist of Class, Individual

Property, Data Valued Property, and Built-in types. Furthermore, the connection mark '^', implication mark '→' and the question mark '?' are introduced as the symbols of SWRL rules [36]. Therefore, the SWRL rules for the Equation (3) can be given as:

| Equation: | $C^E = C^T + C^P$ |
|---|---|
| SWRL: | Energy_pile_system(?Es)^Heat_transfer_tube_cost(?Es,?TubeCost)^Ground_pump_cost(?Es,?PumpCost)^swrlb:add(?EquipmentCost,?TubeCost,?PumpCost)->Equipment_cost(?Es,?EquipmentCost) |

Step 8: The SQWRLQueryTab, embedded in Protégé, can provide a friendly man-machine interface for user to query design solutions by inputting SQWRL rules. A specific instance of cost query for EPS is shown as follows:

| SQWR | Energy_pile_system(?Es)^Heat_transfer_tube_cost(?Es,?tube_cost)^Ground_pump_cost(?Es,?pump_cost)^Equipment_cost(?Es,?equipment_cost)->sqwrl:select(?Es, ?tube_cost, ?pump_cost,?equipment_cost) |
|---|---|

### 2.3. Ontology Verification

For the aim to validate the built ontology model, an ontology verification of the semantic, syntactic correctness, and rules validation is conducted in this section to meet the requirements of multi-objective holistic design for EPS.

(1) Semantic verification

If an ontology model is established by extending existing ones, the ontology comparison techniques are generally employed for semantic validation [35,36]. Since the top-level concepts of the developed OntoEPS follow IFC standards and previous ontologies, each concept is validated semantically by domain experts.

(2) Syntactical verification

By using Pellet reasoner compatible with protégé-OWL 5.2, syntactical validation of the OntoEPS is conducted to detect the errors in the syntax of ontology so as to correct the ontology model accordingly.

(3) Rules verification

The rules validation is completed by running the rules in SWRLTab plug-in. A case study is performed in Section 3 for further function validation of the OntoEPS.

### 3. Case study

#### 3.1. Case Study Description

A case study is performed to validate the OntoEPS system and demonstrate how to design the EPS, holistically, using the developed OntoEPS system. The prototype is a residential building in the Qingdao city of China, as shown in Figure 7. The key parameters of the EPS are listed in Table 1.

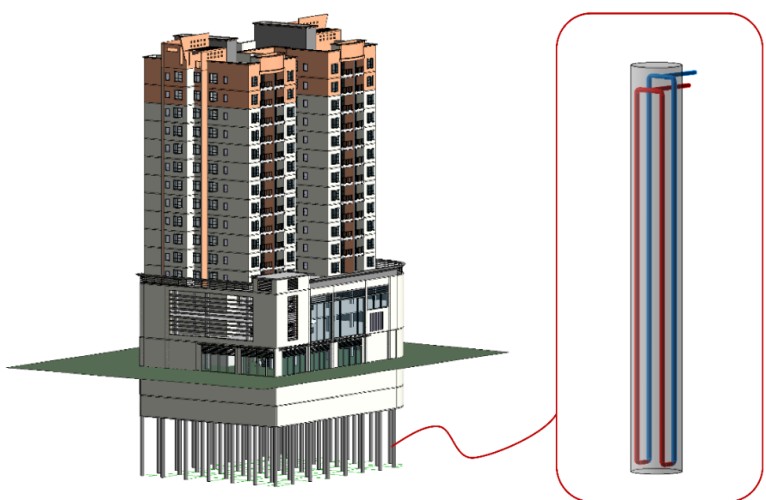

**Figure 7.** 3D BIM model of the prototype.

**Table 1.** Primary design parameters of the energy pile system.

| | |
|---|---|
| The area of heat load | 400,000 m$^2$ |
| Peak of hourly dynamic load | 200 kW |
| The type of the pile | bored pile |
| The number of the pile | 700 |
| The length of the pile | 10 m |
| The type of the heat pump | FSLC 2330H Centrifugal magnetic suspension heat pump |
| The number of the heat pump | 2 |
| The price of the heat pump | 0.175 million (USD) per unit |
| The type of the heat exchanger tube | PE plastic tube |
| The price of the heat exchanger tube | 1 USD/meter |
| The price of the electricity | 0.17 USD/kWh |

Furthermore, five different types of heat exchanger are employed in the prototype building. Due to the lack of field experimental data, the primary parameters of the used heat exchangers are selected according to experience and relevant specifications, as listed in Table 2. Based on a combined consideration of five heat exchanger types and two pile layouts (700 piles and 625 piles), ten design solutions of the EPS are investigated in this case study.

**Table 2.** Primary parameters of heat exchangers.

| Heat Exchanger Type | Length (m) | Heat Exchanged Per Meter (W/m) | Vertical Ultimate Bearing Capacity (kN) |
|---|---|---|---|
| Series double U type | 40 | 25 | 2310 |
| Parallel double U type | 40 | 30 | 2280 |
| Single U type | 20 | 20 | 2300 |
| Single spiral type | 125 | 50 | 2200 |
| Three U type | 60 | 48 | 2250 |

Thereafter, the developed ontology model and preset SWRL rules can be run to generate the new facts of the OntoEPS. The preset SWRL and SQRWL rules are shown in Table 3.

**Table 3.** Preset SWRL and SQWRL rules.

| Rule Number | Rule Type | Specification |
|---|---|---|
| R.1 | Equipment Cost (SWRL) | Calculating heat exchanger costs: $C^T = \sum_{i=1}^{n} C_i^T \times L_i^T \times N_i$<br><br>Energy_pile_system(?Es)^hasPile_type(?Es,?Pile)^pile_type(?Pile)^numbers(?Pile,?N)^hasHeat_transfer_tube_type(?Pile,?Tube)^Heat_transfer_tube_type(?Tube)^Length(?Tube,?L)^Cost(?Tube,?C)^swrlb:multiply(?TubeCost,?L,?C,?N)->Heat_transfer_tube_cost(?Es, ?TubeCost) |
| R.2 | | Calculating pump costs: $C^P = \sum_{j=1}^{m} C_j^P \times N_j^P$<br><br>Energy_pile_system(?Es)^hasGround_pump_type(?Es,?P)^Ground_pump_type(?G)^Cost(?G,?C)^numbers(?G,?N)^swrlb:multiply(?PumpCost,?C,?N)->Ground_pump_cost(?Es, ?PumpCost) |
| R.3 | | Calculating equipment cost: $C^E = C^T + C^P$<br>Energy_pile_system(?Es)^Heat_transfer_tube_cost(?Es,?TubeCost)^Ground_pump_cost(?Es,?PumpCost)^swrlb:add(?EquipmentCost,?TubeCost,?PumpCost)->Equipment_cost(?Es, ?EquipmentCost) |
| R.4 | CostRecovery Period (SWRL) | Calculating the work of heat exchanged by the system per hour:<br><br>$$H^T = \frac{\sum_{i=1}^{n} H_i \times L_i^S \times N_i}{1000}$$<br>Energy_pile_system(?Es)^hasPile_type(?Es,?pile)^pile_type(?pile)^Heat_exchange_per_meter(?pile,?heat)^Length(?pile,?L)^numbers(?pile,?N)^swrlb:multiply(?total_exchange_heat,?heat,?L,?N,0.001) -> Total_exchange_heat(?Es, ?total_exchange_heat) |
| R.5 | | Calculating the cost recovery period: $T^C = \frac{C^E}{H^T \times P}$<br>Energy_pile_system(?Es)^Equipment_cost(?Es, ?equipment_cost)^Electricity_price(?Es, ?electricity_price)^Total_exchange_heat(?Es,?total_exchange_heat)^swrlb:multiply(?x,?electricity_price, ?total_exchange_heat) ^ swrlb:divide(?y, ?equipment_cost, ?x) -> Cost_recovery_time(?Es, ?y) |
| R.6 | CO$_2$ emissionsreduction during cost recovery period (SWRL) | Calculating CO$_2$ emissions reduction during cost recovery period:<br>$CO_2^E = H^T \times T^C \times 0.997$<br>Energy_pile_system(?Es)^Co2(?Es,?co2)^Total_exchange_heat(?Es,?heat)^Cost_recovery_time(?Es,?time) ^swrlb:multiply(?total_co2, ?co2, ?heat, ?time) -> TotalCo2(?Es, ?total_co2) |
| R.7 | Vertical bearing capacity of EPS (SWRL) | Calculating vertical bearing capacity of energy pile system: $Q = \sum_{i=1}^{n} \frac{Q_i^{uk}}{K} \times N_i$<br><br>Energy_pile_system(?Es)^hasPile_type(?Es,?pile)^pile_type(?pile)^Qvk(?pile,?qvk)^K(?pile,?k)^numbers(?pile,?N)^swrlb:divide(?ra,?qvk,?k)^swrlb:multiply(?total_bearing,?ra,?N)->Total_vertical_bearing_capacity(?Es, ?total_bearing) |
| R.8 | Querying rules (SQWRL) | Energy_pile_system(?Es)^Equipment_cost(?Es,?equipment_cost)^Total_exchange_heat(?Es, ?total_exchange_heat)^Cost_recovery_time(?Es,?time)^TotalCo2(?Es,?totalco2)^Total_vertical_bearing_capacity(?Es,?bearing_capacity)->sqwrl:select(?Es, ?equipment_cost,?time, ?totalco2,?bearing_capacity) |

The preset SWRL rules, for the design of economy, safety, and sustainability of EPS, refer to R.1–R.7 in Table 3. After running the preset SWRL rules, the OntoEPS system generates the inferred facts, as shown in Figure 8. Users can also employ SQWRLQueryTab plug-in to inquire the facts for the holistic decisions of design. For example, the SQWRL rules for querying the equipment cost, cost recovery period, CO$_2$ emission reduction, and vertical bearing capacity refer to R.8 in Table 3. The querying results are shown in Figures 8–10.

| Es | total _exchange_heat | Equipment_cost | time | totalco2 | bearing_capacity |
|---|---|---|---|---|---|
| autogen1:Es_parallel_double_u_type | 210.0 | "2,268,000.0" ^^xsd:float | $1.08 \times 10^4$ | 2,261,196.0 | "798,000.0" ^^xsd:float |
| autogen1:Es_parallel_double_u_type_⋯ | 187.5 | "2,250,000.0" ^^xsd:float | $1.2 \times 10^4$ | 2,243,250.0 | "712,500.0" ^^xsd:float |
| autogen1:Es_single_spiral_type_625 | 312.5 | "2,568,750.0" ^^xsd:float | $8.22 \times 10^4$ | 2,561,043.7 | "687,500.0" ^^xsd:float |
| autogen1:Es_single_u_type | 140.0 | "2,184,000.0" ^^xsd:float | $1.56 \times 10^4$ | 2,177,448.0 | "805,000.0" ^^xsd:float |
| autogen1:Es_single_u_type_625 | 125.0 | "2,175,000.0" ^^xsd:float | $1.74 \times 10^4$ | 2,168,475.0 | "787,500.0" ^^xsd:float |
| autogen1:Es_three_u_type | 336.0 | "2,352,000.0" ^^xsd:float | $7 \times 10^3$ | 2,344,944.0 | "787,500.0" ^^xsd:float |
| autogen1:Es_series_double_u_type | 175.0 | "2,268,000.0" ^^xsd:float | $1.296 \times 10^4$ | 2,261,196.0 | "808,500.0" ^^xsd:float |
| autogen1:Es_series_double_u_type_625 | 156.3 | "2,250,000.0" ^^xsd:float | $1.44 \times 10^4$ | 2,243,250.0 | "721,875.0" ^^xsd:float |
| autogen1:Es_single_spiral_type | 350.0 | "2,625,000.0" ^^xsd:float | $7.5 \times 10^3$ | 2,617,125.0 | "770,000.0" ^^xsd:float |
| autogen1:Es_three_u_type_625 | 300.0 | "2,325,000.0" ^^xsd:float | $7.75 \times 10^3$ | 2,318,025.0 | "703,125.0" ^^xsd:float |

**Figure 8.** Execution and results of querying R.8 in Table 3.

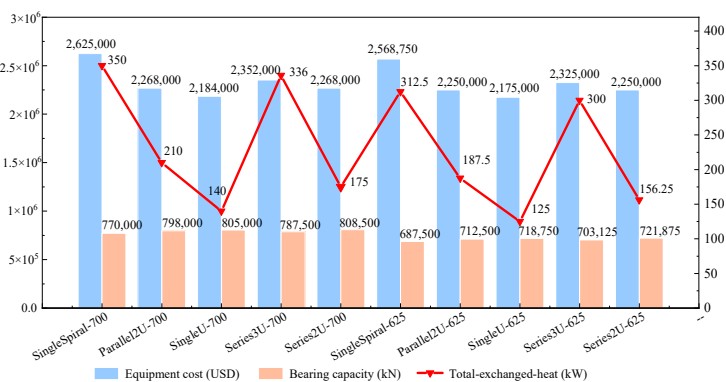

**Figure 9.** Querying results of equipment cost and vertical bearing capacity.

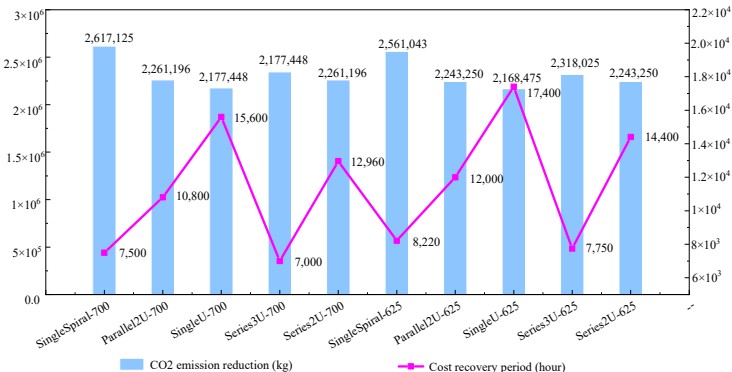

**Figure 10.** Querying results of $CO_2$ emission reduction and cost recovery period.

### 3.2. Applications

This section mainly presents the applications of the ontology model OntoEPS, developed in the case study, and related indicators are also tested based on corresponding SQWRL queries. The SQWRL rules used in this section are shown in Table 4.

**Table 4.** SQWRL rules to query related indicators in the case study.

| Rule Number | Related Indicators | SQWRL Rule |
|---|---|---|
| R.1 | Design solutions with equipment cost less than 0.4 million (USD) | Energy_pile_system(?Es)ˆEquipment_cost(?Es,?equipment_cost)ˆTotal _exchange_heat(?Es, ?total_exchange_heat)ˆCost _recovery_time(?Es,?time)ˆTotalCo2(?Es,?totalco2)ˆTotal_vertical_bearing _capacity(?Es,?bearing_capacity)ˆswrlb:lessThan(?equipment_cost, 2500000)-> sqwrl:select(?Es, ?total_exchange_heat, ?equipment_cost, ?time, ?totalco2, ?bearing_capacity) |
| R.2 | Design solutions with the bearing capacity greater than $7 \times 106$ kN | Energy_pile_system(?Es)ˆEquipment_cost(?Es,?equipment_cost)ˆTotal _exchange_heat(?Es, ?total_exchange_heat)ˆCost _recovery_time(?Es,?time)ˆTotalCo2(?Es,?totalco2)ˆTotal_vertical_bearing _capacity(?Es,?bearing_capacity)ˆswrlb:greaterThan(?bearing_capacity, 700000)-> sqwrl:select(?Es, ?total_exchange_heat, ?equipment_cost, ?time, ?totalco2, ?bearing_capacity) |
| R.3 | Design solutions with total exchanged heat greater than 200 kW and equipment cost less than 0.4 million (USD) | Energy_pile_system(?Es)ˆEquipment_cost(?Es,?equipment_cost)ˆTotal _exchange_heat(?Es, ?total_exchange_heat)ˆCost _recovery_time(?Es,?time)ˆTotalCo2(?Es,?totalco2)ˆTotal_vertical_bearing _capacity(?Es,?bearing_capacity)ˆswrlb:greaterThan(?total_exchange_heat, 200)ˆ swrlb:lessThan(?equipment_cost,2500000)->sqwrl:select(?Es, ?total_exchange_heat, ?equipment_cost, ?time, ?totalco2, ?bearing_capacity) |

Firstly, the designers can query the significant indicators for the holistic performance of the EPS, corresponding to different building characteristics and cooling/heating requirements. R.1 in Table 4 illustrates how the designer uses the SQWRL rules to query the equipment, which costs less than 0.4 million (USD), to build an EPS. Figure 11 demonstrates the execution and results of querying R.1 in Table 4 related to the filtered design solutions, of which the equipment cost is less than 0.4 million (USD) for building an EPS. Figure 12 shows the execution and results of querying R.2 in Table 4, with regard to the filtered design solutions, of which the bearing capacity is greater than $7 \times 106$ kN for building an EPS. Figure 13 shows the execution and results of querying R.3 in Table 4, corresponding to the three filtered design solutions with the equipment cost less than 0.4 million USD and total exchanged heat greater than 200 kW.

| Es | total _exchange_heat | Equipment_cost | time | totalco2 | bearing_capacity |
|---|---|---|---|---|---|
| autogen1:Es_parallel_double_u_type | 210.0 | "2,268,000.0" ^^xsd:float | $1.08\times10^4$ | 2,261,196.0 | "798,000.0" ^^xsd:float |
| autogen1:Es_parallel_double_u_type_ ⋯ | 187.5 | "2,250,000.0" ^^xsd:float | $1.2\times10^4$ | 2,243,250.0 | "712,500.0" ^^xsd:float |
| autogen1:Es_single_u_type | 140.0 | "2,184,000.0" ^^xsd:float | $1.56\times10^4$ | 2,177,448.0 | "805,000.0" ^^xsd:float |
| autogen1:Es_single_u_type_625 | 125.0 | "2,175,000.0" ^^xsd:float | $1.74\times10^4$ | 2,168,475.0 | "718,750.0" ^^xsd:float |
| autogen1:Es_three_u_type | 336.0 | "2,352,000.0" ^^xsd:float | $7\times10^3$ | 2,344,944.0 | "787,500.0" ^^xsd:float |
| autogen1:Es_series_double_u_type_625 | 156.2 | "2,250,000.0" ^^xsd:float | $1.44\times10^4$ | 2,243,250.0 | "721,875.0" ^^xsd:float |
| autogen1:Es_three_u_type_625 | 300.0 | "2,325,000.0" ^^xsd:float | $7.75\times10^3$ | 2,318,025.0 | "703,125.0" ^^xsd:float |
| autogen1:Es_series_double_u_type | 175.0 | "2,268,000.0" ^^xsd:float | $1.296\times10^4$ | 2,261,196.0 | "808,500.0" ^^xsd:float |

**Figure 11.** Execution and results of querying R.1 in Table 4.

| Es | total _exchange_heat | Equipment_cost | time | totalco2 | bearing_capacity |
|---|---|---|---|---|---|
| autogen1:Es_parallel_double_u_type | 210.0 | "2,268,000.0" ^^xsd:float | $1.08\times10^4$ | 2,261,196.0 | "798,000.0" ^^xsd:float |
| autogen1:Es_parallel_double_u_type_ ⋯ | 187.5 | "2,250,000.0" ^^xsd:float | $1.2\times10^4$ | 2,243,250.0 | "712,500.0" ^^xsd:float |
| autogen1:Es_single_u_type | 140.0 | "2,184,000.0" ^^xsd:float | $1.56\times10^4$ | 2,177,448.0 | "805,000.0" ^^xsd:float |
| autogen1:Es_single_u_type_625 | 125.0 | "2,175,000.0" ^^xsd:float | $1.74\times10^4$ | 2,168,475.0 | "718,750.0" ^^xsd:float |
| autogen1:Es_three_u_type | 336.0 | "2,352,000.0" ^^xsd:float | $7\times10^3$ | 2,344,944.0 | "787,500.0" ^^xsd:float |
| autogen1:Es_series_double_u_type_625 | 156.2 | "2,250,000.0" ^^xsd:float | $1.44\times10^4$ | 2,243,250.0 | "721,875.0" ^^xsd:float |
| autogen1:Es_three_u_type_625 | 300.0 | "2,325,000.0" ^^xsd:float | $7.75\times10^3$ | 2,318,025.0 | "703,125.0" ^^xsd:float |
| autogen1:Es_series_double_u_type | 175.0 | "2,268,000.0" ^^xsd:float | $1.296\times10^4$ | 2,261,196.0 | "808,500.0" ^^xsd:float |
| autogen1:Es_single_spiral_type | 350.0 | "2,625,000.0" ^^xsd:float | $7.5\times10^3$ | 2,617,125.0 | "770,000.0" ^^xsd:float |

**Figure 12.** Execution and results of querying R.2 in Table 4.

| Es | total_exchange_heat | Equipment_cost | time | totalco2 | bearing_capacity |
|---|---|---|---|---|---|
| autogen1:Es_parallel_double_u_type | 210.0 | "2,268,000.0" ^^xsd:float | $1.08 \times 10^4$ | 2,261,196.0 | "798,000.0" ^^xsd:float |
| autogen1:Es_three_u_type_ | 336.0 | "2,352,000.0" ^^xsd:float | $7 \times 10^3$ | 2,344,944.0 | "787,500.0" ^^xsd:float |
| autogen1:Es_three_u_type_625 | 300.0 | "2,325,000.0" ^^xsd:float | $7.75 \times 10^3$ | 2,318,025.0 | "703,125.0" ^^xsd:float |

**Figure 13.** Execution and results of querying R.3 in Table 4.

*3.3. Discussion*

From Figure 11, the calculated cost recovery periods of the eight design solutions are less than the design life (i.e., 20~30 years) of EPS. Among the filtered design solutions, the one with a single U type heat exchanger has the lowest cost, from which the designers can also adjust the other parameters for further optimization in a holistic manner. From Figure 12, the calculated bearing capacity of pile groups for nine design solutions satisfies the assumed bearing capacity limit above. In addition, all the calculated values of bearing capacity based on the OntoEPS are of the same order of magnitude, i.e., the corresponding rules of calculation and query are technically reasonable, with respect to existing research [18]. Furthermore, as two significant indicators of the EPS, the peak of the hourly dynamic load of the EPS calculated by software DEST-C, with the certain equipment cost, can be used to query corresponding design solutions. Specifically, it can be noted from Figure 13 that the design solution with parallel double U exchanger and pile layout of 700 piles involves the least cost. It should be noted that the selection of pump type may lead to significant cost differences within the construction of EPS.

Compared with the single-objective (only considering the main functional design objectives of EPS for loading capacity and heat transfer) design method, the multi-objective design method can consider the cost for life cycle design, construction and maintenance, return of investment, and $CO_2$ emission related sustainable requirements, holistically. Therefore, the multi-objective design method can provide more alternative optimization schemes for engineers in a comprehensive insight. The results of this case study verify that the developed OntoEPS can provide several alternative design schemes for engineers in practice under specific restrictive conditions. Furthermore, users can choose the optimal scheme from these schemes based on different demands. However, there still exists some limitations in this study, as follows:

(1) For the cost, we only consider the cost of the material. The cost of labor and equipment is neglected.
(2) The developed OntoEPS is not fully automated. Users still need to compile and type the corresponding SQWRL rules to achieve a specific function.
(3) The developed OntoEPS is limited in the Protégé software environment. That means the engineers should have basic knowledge of ontology to use this approach in practice.

**4. Conclusions**

In this paper, an ontology-based approach, for multi-objective holistic design of EPS, is presented. A prototypical decision-making system with new ontology framework considering technical, economic, and sustainable aspects is developed. A case study with typical applications is performed as well to illustrate the details on how to leverage knowledge query to validate the practicability and scientific feasibility. Based on IFC standards, validation, including the semantic correctness, the syntactic correctness, and SWRL rules, is conducted by employing the ontology alignment, merging, or comparison techniques. Typical applications of the preset SWRL rules, querying results and inferred facts of the economy, safety, and sustainability of EPS design are illustrated, considering different design parameters, e.g., the life cycle cost, return of investment, bearing capacity, heat exchange, and $CO_2$ emission indicators. The developed OntoEPS and corresponding ontology framework can also be extended to the other geothermal system (e.g., energy tunnel system). For the future work, the knowledge acquisition method involving more semantic elaboration (e.g., the interaction with large-scale numerical analysis) will be the focus.

**Author Contributions:** Conceptualization, K.M. and C.C.; methodology, K.M. and H.L. (Hailong Liu); software, K.M.; validation, C.C.; formal analysis, H.L. (Hailong Liu); investigation, C.C.; resources, C.C.; data curation, K.M.; writing—original draft preparation, K.M.; writing—review and editing, H.L. (Haijiang Li); visualization, K.M.; supervision, C.C. and H.L. (Haijiang Li); project administration, C.C.; funding acquisition, K.M. and C.C. All authors have read and agreed to the published version of the manuscript.

**Funding:** This work is supported by the National Natural Science Foundation of China (Grant No. 52178315, 51878109), the National Science Foundation for Young Scientists of China (Grant No. 52108326), the National Science Fund for Excellent Young Scholars of China (Grant No. 51722801), the Fundamental Research Funds for the Central Universities (Grant No. 3132019601), Cultivation project of Innovation talent for doctorate student (CXXM2019BS008 and BSCXXM022).

**Institutional Review Board Statement:** Not applicable.

**Informed Consent Statement:** Not applicable.

**Data Availability Statement:** The data used to support the findings of this study are available from the corresponding author upon request.

**Conflicts of Interest:** The authors declare no conflict of interest.

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
