# Peer review of "Ontology-Based Approach Supporting Multi-Objective Holistic Decision Making for Energy Pile System"

_buildings, doi:10.3390/buildings12020236_

Round 1

Reviewer 1 Report

The authors propose an ontology-based approach of leverage knowledge for modeling energy pile system (EPS) design to achieve multi-objective design optimization. Also, this work presents a prototypical ontology-based decision tool for optimizing the EPS. 

The manuscript must be strongly revised and improved before it is considered for publication in the journal.

Comments:

I recommend that the authors show the difference between this work and the work published with DOI: 10.1109/ACCESS.2020.2986229.

The main weakness of this work is related to the correct definition of multi-objective (MO) formulation. It looks like that the authors define a MO, but there doesn't exist a correct definition of the main components properly of a mathematical optimization model, and indeed (e.g., parameters, decision variables, objective functions, etc.). Please, a correct definition of MO formulation must be well described.

The references sections must be extensively improved. Please, review the information for each article and establish the date of consultation for website references (e.g., remove all [J]; remove ISSN number, ref: 6; remove all 'et al.' and put all authors; remove 'ARTICLE', ref: 18; remove all '//', ref: 32,...; review ref: 35)

Author Response

We thank very much the reviewer for the valuable time reviewing our work and the insightful comments that helped to improve the quality of the paper. We have adequately considered all comments, carefully made corrections and modifications which are highlighted in yellow in the revised manuscript. Please check them. And we will follow the reviewer's further comments to revise the manuscript if you have.

Details refer to the attachment.

Reviewer 2 Report

The paper presents a new multi-objective approach for designing energy pile system of heat pumps. Nevertheless, the paper has some drawbacks that should be addressed to be published:

  1. Introduction section should end with a summary of each section of the paper working as a reading guide of the paper.
  2. It would be clarifying if there is a deeper discussion on the main advantages of using multi-objective optimization instead of using single-objective optimization.
  3. There are some acronyms and abbreviations that are not defined, such as SWRL, ORL, etc., that needs to be defined in the first time they are used and then the acronym should be used, eg,“…Semantic Web Rule Language (SWRL)…”
  4. The section with the mathematical formulation of the problem needs to be improved in what regards to small typos and extra spaces near the formulae.
  5. In what regards to units used, the kWh should be used without the space between “kW” and “h” (kW h).
  6. The quality of the images should be improved, eg, Figure 3, etc.
  7. Figures 5, 6, 7, 8, 10, 13, 14 and 15 are very blurry and, in my opinion some of them are dispensable. Please verify which ones are essential for the paper and use adequate image sizes and resolutions.
  8. Please try to combine Tables 3 to 10 into one multi-column and multi-row table with essential information
  9. Despite using RMB as currency, it is suggested to provide a conversion factor to US dollar and/or to EUR.
  10. Despite the very detailed description of the methods and case study, the results are not properly presented and explored. Moreover, a sound and critically discussion analysis on the results obtained is required.

Author Response

(The authors gave the same response as above.)

Reviewer 3 Report

This paper presents a multi-objective holistic approach for energy pile system design. The system uses ontology based multi-domain knowledge orchestration, with and eleven factors model regarding financial, safety and environmental impact, for early design stage. The manuscript is well written and the methods described quite clearly, with few exceptions. I believe it can be further improved with minor revision. Here are some suggestions:

The authors should clearly express the aim of this work and assess its novelty concerning the existing literature, which consists of several papers published on this topic, as also proved by the number of references in the manuscript.

Please consider shortening the title and make it clearer. Shorter sentences could help.

Introduction: Please, add a description of the structure of the manuscript at the end of the introduction section and the methodology followed.

General remarks on the model: in the discussione, the authors should better explain which are the advantages provided by the holistic approach they are proposing, compared to the systems mentioned in the literature review. Moreover, they should also define the applicability of this model and its limitations.

Author Response

(The authors gave the same response as above.)

Round 2

Reviewer 1 Report

The authors have answered all the recommendations. In my opinion, the article must be considered for publication in the journal.

Reviewer 2 Report

The paper has been improved. Therefore, in my opinion it should be considered for publication.

This manuscript is a resubmission of an earlier submission. The following is a list of the peer review reports and author responses from that submission.